# Polystyrene-Fe_3_O_4_-MWCNTs Nanocomposites for Toluene Removal from Water

**DOI:** 10.3390/ma14195503

**Published:** 2021-09-23

**Authors:** Thamer Adnan Abdullah, Tatjána Juzsakova, Rashed Taleb Rasheed, Ali Dawood Salman, Viktor Sebestyen, Endre Domokos, Brindusa Sluser, Igor Cretescu

**Affiliations:** 1Sustainability Solutions Research Laboratory, Faculty of Engineering, University of Pannonia, 8200 Veszprém, Hungary; thamer.abdullah@mk.uni-pannon.hu (T.A.A.); yuzhakova@almos.uni-pannon.hu (T.J.); ali.dawood@mk.uni-pannon.hu (A.D.S.); sebestyen.viktor88@gmail.com (V.S.); domokose@uni-pannon.hu (E.D.); 2Chemistry Branch, Applied Sciences Department, University of Technology, Baghdad 10001, Iraq; 100010@uotechnology.edu.iq; 3Faculty Chemical Engineering and Environmental Protection, “Gheorghe Asachi” Technical University of Iasi, 73, Blvd. D. Mangeron, 700050 Iasi, Romania

**Keywords:** nanocomposites, toluene removal, water treatment, nanomaterials, polystyrene, magnetite

## Abstract

In this research, multi-walled carbon nanotubes (MWCNTs) were functionalized by oxidation with strong acids HNO_3_, H_2_SO_4_, and H_2_O_2_. Then, magnetite/MWCNTs nanocomposites were prepared and polystyrene was added to prepare polystyrene/MWCNTs/magnetite (PS:MWCNTs:Fe) nanocomposites. The magnetic property of the prepared nano-adsorbent PS:MWCNTs:Fe was successfully checked. For characterization, X-ray diffraction (XRD), scanning electron microscopy (SEM), transmission electron microscopy (TEM), Fourier transform infrared spectroscopy (FTIR), Raman spectroscopy, and BET surface area were used to determine the structure, morphology, chemical nature, functional groups, and surface area with pore volume of the prepared nano-adsorbents. The adsorption procedures were carried out for fresh MWCNTs, oxidized MWCNTs, MWCNTs-Fe, and PS:MWCNTs:Fe nanocomposites in batch experiments. Toluene standard was used to develop the calibration curve. The results of toluene adsorption experiments exhibited that the PS:MWCNTs:Fe nonabsorbent achieved the highest removal efficiency and adsorption capacity of toluene removal. The optimum parameters for toluene removal from water were found to be 60 min, 2 mg nano-sorbent dose, pH of 5, solution temperature of 35 °C at 50 mL volume, toluene concentration of 50 mg/L, and shaking speed of 240 rpm. The adsorption kinetic study of toluene followed the pseudo-second-order kinetics, with the best correlation (R^2^) value of 0.998, while the equilibrium adsorption study showed that the Langmuir isotherm was obeyed, which suggested that the adsorption is a monolayer and homogenous.

## 1. Introduction

Though water makes up three-quarters of the world, the availability of water that is safe enough for human consumption is less than 1%, which leaves a large population living with inadequate drinking water [1]. Naturally available fresh water is regularly contaminated by a number of anthropogenic activities and industrial processes. The significant increase in population growth has led to enormous industrial applications, which results in the release of organic pollutants, especially from manufacturing industries [2]. The most frequently encountered organic pollutants include toluene, dyes, and oil spills. Toluene is used as a manufacturing component to produce polymer, rubber, medicine, dyes, inks, benzene, and explosives. Byproducts of these productions are mainly discharged into the atmosphere and have severely polluted the water. The consistent release of toluene into the atmosphere and surface water has equally negatively impacted the health of humans, animals, and the aquatic system. The major health hazards due to the exposure of toluene include central nervous system (CNS) dysfunction and narcosis [3]. Additionally, due to the persistent contact with water, toluene forms toxic compounds that are injurious to human health [4].

The common existing treatment procedures for depollution of contaminated water include electrochemical oxidation, biological processes, adsorption, photocatalysis, gravity separation, etc. [5,6]. These techniques usually suffer from limitations, such as poor separation efficiency, time-consuming, and higher energy costs, and sometimes lead to secondary pollution, i.e., generating waste [7]. Similarly, there are a number of membrane techniques used for oil–water separation, including reverse osmosis, nano-filtration, and ultra-filtration [8]. Though membrane technology is highly efficient and has low energy consumption, they are not suitable for oil–water separation because they require extra energy, chemicals, and funds for cleaning the membrane [9]. The situation has motivated quite a large number of researchers to develop efficient and alternative methods that are environmentally friendly.

The applications of nanotechnology have brought a considerable revolution in materials science, and researchers have flocked to it because of the impeccable properties shown by nanomaterials, such as higher sorption rate, super-hydrophobicity, super-oleophilicity, and greater mechanical strength [10]. Multi-walled carbon nanotubes (MWCNTs), compared with other nanomaterials, have shown a relatively higher affinity of adsorption for removal of volatile organic compounds, as reported by Parmar et al. [11], for the removal of heavy metal ions [12] and for the degradation of dyes [13]. Likewise, nowadays, MWCNTs are considered as potential adsorbents for numerous remediation applications, especially in the environmental field, including removal of organic pollutants and heavy metals from aqueous media [14,15]. The synthesis methods for carbon nanotubes’ (CNTs) preparation are usually selected in light of the required suitable properties and the specific field of application where these synthesized nanomaterials will be used.

The use of nanocomposites in removing organic pollutants such as toluene is more preferable to other similar methods, including membrane separation, in one way or another [16]. This is attributed to a number of characteristic features possessed by these nanocomposites, as narrated by Hu et al., such as low density, electrical conductivity, large specific surface area, high inherent strength, higher adsorption capacity, good hydrophobicity, thermal and chemical stability, high aspect ratio, fast adsorption rate, oleophilic characteristics, and hydrogen storage capacity [17]. These properties have been confirmed by Wang et al. [18].

Tan et al. have successfully developed an environmentally friendly adsorbent on raw corn straw after deposition of SiO_2_/ZnO nanocomposite particles, with excellent super-oleophilic and super-hydrophobic characteristics [19]. Cao et al., based on their research, have reported that the titania/carbon nanotube composite (TiO_2_/CNT) has shown enhanced absorption for the removal of organic pollutants in comparison to pure TiO_2_ nanoparticles [20]. Similarly, Sobhanardakani and Zandipak, in 2018, reported on functionalized silica-coated magnetite nanocomposites for removing organic pollutants from a water solution [21]. Kirti et al. exploited the biomass functionality in iron nanocomposites for the potential removal of four dyes, which included both anionic and cationic dyes [22]. Parangusan et al. reported the fabrication of membrane oil absorbents based on carbon nanotube-reinforced polystyrene nanocomposites by an electrospinning technique [23]. Rekos et al. prepared magnetic nanocomposite adsorbents by impregnating graphene oxide with three different polymers, i.e., chitosan, polystyrene, and polyaniline, for the removal of a typical endocrine disruptor, bisphenol-A, from aqueous solutions [24]. A new adsorbent based on silica gel impregnated with deep eutectic solvents (DESs) was prepared to increase the adsorption efficiency of toluene and other organic compounds [25].

In this article, MWCNTs were treated with strong acids for the oxidation process (ox-MWCNTs), at first by strong acids (HNO_3_, H_2_SO_4_) followed by H_2_O_2_. Two solutions, Fe^+3^:FeCl_3_.6H_2_O and Fe^+2^:FeSO_4_.7H_2_O, were used to prepare magnetite (Fe_3_O_4_) over ox-MWCNTs for the formation of magnetite/MWCNTs (MWCNTs-Fe) nanocomposites. Then, polystyrene (PS) was added to the mixture in order to prepare the final polystyrene:multi-walled carbon nanotubes:magnetite (PS:MWCNTs:Fe) nanocomposites with a PS to MWCNTs:Fe weight (wt) ratio of 1:3. Many researchers have prepared polymer:MWCNTs or magnetite for water treatment [25,26,27,28,29,30]. Researchers used to add a small amount of CNT-based polymer to prepare the superhydrophobic sorbent. In our work, we prepared a nano-sorbent based on CNTs, and we successfully improved their sorption ability for toluene removal. Two references were added. The removal of toluene from water by adsorption was carried out for fresh MWCNTs, ox-MWCNTs, MWCNTs-Fe, and PS:MWCNTs:Fe. The results of the PS:MWCNTs:Fe nanocomposites indicated the highest adsorption capacity and the highest removal efficiency of toluene compared with other Fe-MWCNTs and ox-MWCNTs nanocomposites.

## 2. Materials and Methods

The chemicals and standard samples that were used for the current research are provided in this section with brief and specific details. Nitric acid (HNO_3_, 99%; from Merck Chemicals Co. Budapest, Hungary), polystyrene (99%; from Merck Chemicals Co. Budapest, Hungary), cetyl-trimethyl-ammonium bromide (CTAB, C_19_H_42_BrN, 99%; from Merck Chemicals Co. Budapest, Hungary), iron tetrachloride (Fe^+3^: FeCl_3_.6H_2_O; 99%; from Merck Chemicals Co. Budapest, Hungary), iron sulfate (Fe^+2^: FeSO_4_.7H_2_O; 99.7%; Merck Chemicals Co. Budapest, Hungary), hydrochloric acid (HCl; 99.7%; VWR Chemicals BDH Co.Debrecen, Hungary), sodium hydroxide (NaOH; 99%; VWR Chemicals BDH Co. Debrecen, Hungary), and commercial grade MWCNTs samples (TNNF-6 type; Times Nano China, made by the chemical vapor deposition technique (CVD)) were used.

### 2.1. Synthesis of Polystyrene:MWCNTs:Magnetite Nanocomposites (PS:MWCNTs:Fe)

Fresh MWCNTs were oxidized by adding two strong acids, i.e., H_2_SO_4_ and HNO_3_, in a ratio of 3:1, respectively. The solution was subjected to ultrasonication for 6 h followed by MWCNTs washing, and the solution was filtered. After that, H_2_O_2_ was added, and the solution was again ultrasonicated for the second oxidation process to make sure that the MWCNTs were completely oxidized (ox-MWCNTs) and the carboxyl group was generated [31,32]. Then, the ox-MWCNTs were washed until a pH near 7 was achieved. The ox-MWCNTs were dried at 90 °C overnight. Fe_3_O_4_/MWCNTs (MWCNTs-Fe) nanocomposites were prepared by using the following process: Two solutions, Fe^+3^:FeCl_3_.6H_2_O and Fe^+2^:FeSO_4_.7H_2_O, having a molar ratio of 2:1 respectively, were mixed. In order to avoid the agglomeration of magnetite formation, CTAB was added to the mixture of iron salts. MWCNTs were added to the mixture along with continuous stirring at 40 °C until the solution became the same as clay. Nitrogen gas was passed during the process to prevent the oxidation of Fe^+2^ to Fe^+3^. An ammonium hydroxide solution was added to the mixture dropwise, and the addition was stopped when the pH of the solution was 10. The final MWCNTs-Fe were washed and dried at 50 °C using a vacuum oven for 12 h [33].

Chloroform was used for the dissolution of polystyrene. At first, 100 mg of Fe-MWCNTs/L of chloroform was sonicated at 50 °C for 1 h. Then, polystyrene was added to the mixture in a 1:3 PS:Fe-MWCNTs weight ratio. The mixture was sonicated for a further 3 h. The mixture was then mixed with a magnetic stirrer for 24 h at 40 °C using a hot plate [34]. The final PS:MWCNTs:Fe nanocomposites were formed. The sample was separated, washed, and then dried at 50 °C overnight. The magnetic property was successfully achieved in the prepared nano-sorbent (PS:MWCNTs:Fe). Figure 1 presents the schematic flow chart for the preparation of the PS:MWCNTs:Fe nanocomposites.

### 2.2. Adsorption Experimental Work

Adsorption experiments were carried out in a batch process. The stock solution was first prepared for the adsorption tests. Standard toluene was used to prepare the solution with a concentration of 200 mg/L in deionized distilled water. The stock solution was ultrasonicated for 1 h and then mixed for 24 h at ambient temperature. Before each batch experiment, the solution was sonicated for 30 min. The batch volume was 50 mL, taken in a 100 mL glass conical flask, and then 2 mg of adsorbent was added. After adding the nano-sorbent, the conical flask containing the solution was transferred to a shaker with a speed of 240 rpm. A number of important parameters were measured experimentally, including the range of time, different sorbent weights, different solution temperatures, and different pH values. The magnetic field was successfully used for the separation process [33,34]. A magnet was used after each batch experiment for nano-sorbent separation. A schematic figure of the oil removal process can be seen in Figure 2.

High-performance liquid chromatography (HPLC) (Knauer, Germany System) (C 18, Bond pack 3 μm, (25 cm, 4.6 mm)) was used to measure the concentration of toluene after each experiment. The toluene stock solution was used as a blank without the addition of the nano-sorbent, to ensure that the loss of toluene concentration is not related to the volatility of toluene or the adsorption of toluene on the wall of the conical glass. The toluene removal efficiency and adsorption capacity were calculated using Equations (1) and (2) [35]:Percentage removal (%) = [(Co − Ct)/Co] × 100(1)
where Co = initial toluene concentration, mg/L, and Ct = final kerosene concentration, mg/L.
qt = (Co − Ct)V/W(2)
where V = volume of solution, L, W = weight of prepared metal oxides, g, and qt = final adsorption of toluene.

In order to achieve a calibration standard curve for toluene, the standard toluene solutions were prepared at different concentrations, i.e., 10, 25, 50, and 100 ppm. Toluene concentrations were found using HPLC. The toluene standard calibration curve is shown in Figure 3, which is plotted as the function of toluene concentrations against the area under the peak achieved through HPLC.

## 3. Results and Discussion

### 3.1. Characterization Results

#### 3.1.1. Fourier Transform Infrared (FTIR) Spectroscopy

Figure 4 shows the FTIR spectra for all prepared samples. The 3425 and 1625 cm^−1^ bands belong to water, while the bending vibrations at 2925, 2845, 1460, and 1395 cm^−1^ are attributed to the C–H stretching mode of the aromatic ring, C–H stretching in the branches, C=C aromatic ring stretching, and C–C stretching for the MWCNTs, respectively. After acid treatment, MWCNTs (ox-MWCNTs) and the generation of a carboxyl group could have resulted in the formation of bands at 3420, 1490, and 1400 cm^−1^ related to the stretching vibration of the O–H group in carboxylic acid and the vibration of the C=O carboxyl and C–O stretching bond of free –COOH groups, respectively. The improvement in the peak height at 720 cm^−1^ for Fe-MWCNTs in Figure 4 confirms that the metal oxide (Fe–O) was bonded onto oxygen in the carboxylic groups through the –COO– magnetite bonds. After the adsorption process of toluene using PS:MWCNTs:Fe in Figure 4, the improved peaks at 2925 and 2845 cm^−1^ are indicative of the C-H stretching mode of the aromatic ring and the C–H stretching in the –CH_3_ in toluene, respectively. The FTIR bands appearing at 1035 and 545 cm^−1^ are attributed to in-plane and out-of-plane C–H bending, respectively [23,34,35,36]. Table 1 shows the peak assignments and type of vibration for all prepared sorbents [37].

#### 3.1.2. X-ray Diffraction Investigations

X-ray diffraction (XRD) spectra provide information about the structure of MWCNTs, as presented in Figure 5 (the diffraction patterns). In the case of fresh MWCNTs, they show the main peaks at (002), (100), and (101), appearing at 25.9, 43, and 44°, respectively [35,38,39]. There is no change in the structure of the MWCNTs after oxidation using strong acids, as can be seen in Figure 5. When Fe_3_O_4_ is added to the MWCNTs, the main peaks at (220), (311), (400), (511), and (440) correspond to 31.42, 44.53, 53.35, 57.01, and 63.11°, respectively [40,41]. The XRD patterns for PS:MWCNTs:Fe nanocomposites show a very small intensity peak at 19.4°, which belong to a very low polystyrene content [36], in comparison with MWCNTs and Fe_3_O_4_.

#### 3.1.3. Raman Spectroscopy Investigations

Raman spectroscopy analysis was used to characterize the fresh MWCNTs, ox-MWCNTs, Fe-MWCNTs, PS:MWCNTs:Fe, and toluene adsorption-MWCNTs. There are three sharp peaks at 1338, 1580, and 2635 cm^−1^ in all samples, which correspond to the disorder-induced band (D band), the Raman allowed tangential mode (G band), and the second-order harmonic band G’ (Graphite) band, attributed to the overtone of the D and G bands, respectively [42,43,44]. The Raman spectrum shows a second-order harmonic band G’ (Graphite) at 2635 cm^−1^ corresponding to the D + G bands. In the case of oxidized MWCNTs (ox-MWCNTs) and then modified with magnetite (Fe-MWCNTs), which is represented in Figure 6, the D, G, and G’ were changed, and then after adding polystyrene to the Fe-MWCNTs and PS:MWCNTs:Fe nanocomposites formed, the D, G, and G’ bands were significantly improved. This is related to bonding of the polymer in Fe-MWCNTs. The Raman spectrum was performed for PS:MWCNTs:Fe nanocomposites after toluene adsorption, as can be seen in Figure 6, where the enhancement of peaks’ resolution leads to another band issued at 4150 cm^−1^ that refers to toluene adsorption on the PS:MWCNTs:Fe nanocomposites.

#### 3.1.4. Scanning and Transmission Electron Microscopy Investigations

The scanning electron microscopy (SEM) images for the prepared nanocomposites are shown in Figure 7. Taking into consideration that the magnetite nanoparticles were prepared over MWCNTs in a ratio of 1:1, it was pointed out according to Figure 7a,b that Fe_3_O_4_ particles prepared over MWCNTs were well within the nanoscale, and the shapes were found to be nanotubes; hence, it is clear that there are no effects on the shape of the carbon nanotubes during oxidation of the MWCNTs with a strong acid and even after the magnetite/MWCNTs nanocomposites’ preparation. After modification of MWCNTs/magnetite with the polymer, PS:MWCNTs:Fe has been very clearly shown in Figure 7c,d, pointing out that the mixing of polystyrene during the nanocomposites’ preparation helps the agglomeration of the MWCNTs and magnetite particles. The TEM images in Figure 8a,b show MWCNTs-magnetite nanocomposites. The images indicate that there is no effect on the tubular shape of the CNTs during the preparation and when the magnetite nanoparticles appear over the MWCNTs. Figure 8b shows that the magnetite nanoparticle diameter is less than 10 nm. Figure 8c,d show the PS:MWCNTs:Fe nanocomposites, from which it was also observed that the agglomerations of the magnetite particles occur after the addition of polystyrene. Similarly, the tubular shape of the CNTs is still not affected even after the process of the nanocomposites’ preparation is completed. Energy dispersive X-ray (EDX) analysis was carried out for Fe-MWCNTs. Figure 8e shows the presence of magnetite (Fe_3_O_4_) over the MWCNTs, taking into consideration the presence of the iron, oxygen, and carbon atoms, as can be seen from EDX analysis.

#### 3.1.5. Determination of Surface Area and Pore Volume

The surface area, pore volume, and average pore size values of fresh, oxidized, and modified MWCNTs samples were determined by the Brunauer, Emmett, and Teller (BET) technique, as presented in Table 2. Fresh MWCNTs showed a total surface area of 161 m^2^/g. Ox-MWCNTs could form carboxyl and hydroxy functional groups, which might block the micropore openings [45] and also result in a small decrease in the BET surface area from 161 to 146 m^2^/g. Table 2 indicates that the surface and volume micro-values of fresh MWCNTs decreased after MWCNTs oxidation with strong acids. Fe_3_O_4_ with a particle size less than 10 nm was confirmed through the SEM and TEM results. The deposition of magnetite to MWCNTs in a weight ratio of 1:1 leads to an increase in the surface area of the support. The total surface area of the Fe_3_O_4_/MWCNTs (233 m^2^/g) was higher than the oxidized MWCNTs, indicating that the magnetite is dispersed and incorporated onto the surface of the MWCNTs, and also caused blocking of the micropores of the MWCNTs. Meanwhile, adding polystyrene to the Fe_3_O_4_/MWCNTs sample significantly reduced the surface area from 233 to 136 m^2^/g, and the pore volumes were also reduced. The weight ratio of the polystyrene:Fe_3_O_4_/MWCNTs sample was 1:3. Polystyrene incorporation caused the agglomeration of the Fe_3_O_4_/MWCNTs particles, as confirmed by SEM and TEM. A small decrease in the surface area from 136 to 129 m^2^/g was observed after the toluene adsorption test (Table 2). This supports the fact that some adsorbed hydrocarbon molecules remained in the bulk of the sample after the adsorption test and drying at 105 °C under vacuum.

### 3.2. Adsorption Investigations

#### 3.2.1. Effect of Contact Time

Different contact times were used (15, 30, 60, and 120 min) to study the removal of toluene using fresh MWCNTs, Fe-MWCNTs, and PS:MWCNTs:Fe nano-sorbents. The initial toluene concentration was 50 mg/L in all batch experiments, the solution volume was 50 mL, and the weight of the samples was 2 mg. Figure 9a,b show the removal efficiency and decrease in the concentration of toluene during the process time. The PS:MWCNTs:Fe nano-sorbent achieved the highest removal efficiency and the highest decrease in toluene concentration after 60 min as compared with fresh MWCNTs and Fe-MWCNTs.

The highest removal efficiency of 62% was achieved using PS:MWCNTs:Fe, which also exhibited the highest adsorption capacity of 1072 mg/g, as can be seen in Figure 10a,b.

#### 3.2.2. Effect of Sorbent Dose

The effect of different adsorbent doses was studied using PS:MWCNTs:Fe nanocomposites. A range of adsorbent doses (1, 2, 4, and 6 mg of PS:MWCNTs:Fe nanocomposites) were used to study the removal efficiency of toluene from water. The following parameters were used: 60 min at ambient temperature, 50 mL solution volume, and 50 mg/L toluene concentration. The results indicated that a 2 mg dose of PS:MWCNTs:Fe nanocomposite exhibited the best removal efficiency of toluene, as shown in Figure 11a.

#### 3.2.3. Effect of Solution pH

The effect of pH on the removal of toluene from water was studied. Different pH levels (2, 5, 7, 8, and 10) over the PS:MWCNTs:Fe sample were used with the following parameters: time was 60 min, 2 mg adsorbent dose, 240 rpm shaking speed, 50 mg/L toluene concentration, and 50 mL solution volume at ambient temperature. The results showed that at a pH of 5, the highest adsorption of toluene over PS:MWCNTs:Fe was achieved, as shown in Figure 11b.

#### 3.2.4. Effect of the Solution Temperature

The effect of temperature on the toluene removal efficiency was studied for the temperatures of 15, 25, 35, and 45 °C. The PS:MWCNTs:Fe sample was used for the measurement. The reaction time was 60 min, the weight of the adsorbent was 2 mg, the shaking speed was 240 rpm, the kerosene concentration was 50 mg/L, and the volume of the solution was 50 mL. The results showed that 35 °C was the optimum temperature for toluene removal from water over PS:MWCNTs:Fe as it demonstrated the highest efficiency, as shown in Figure 11c.

Figure 11d shows the toluene concentrations of 50, 100, 200, and 300 ppm, which were used to study the kinetic behavior of toluene adsorption over the nanocomposite adsorbent. Table 3 summarizes the data of different oil adsorption capacities reported by different researchers that were available in the literature, which are compared with the PS:MWCNTs:Fe nanocomposite. The data also confirmed that the PS:MWCNTs:Fe nanocomposite can be effectively used for the adsorption of toluene.

### 3.3. Kinetic Investigations

The kinetic adsorption study was carried out to determine the rate of removal of toluene from water. The kinetic models are instrumental in the determination of the rate of removal at which the adsorbent efficiently removed the adsorbate [58]. In order to find out the rate of removal of toluene, the obtained experimental adsorption data were analyzed using pseudo-first-order, pseudo-second-order, and Weber–Morris kinetic models, followed by the Langmuir and Freundlich isotherm [59]. Figure 12a,b show the resulting graphs from the kinetic models for toluene adsorption. The graph shown in Figure 12a confirmed that the pseudo-second-order model was the best fitting model, with the highest correlation coefficient (R^2^) value of 0.998. Equation (3) was used for the calculation of the second-order rate constant:(3)$${\frac{t}{q}}_{t}=\;\frac{1}{K^{2}\times q_{e}^{2}}+\;\frac{t}{q_{e}}$$ where *k*^2^ is the second-order rate constant, while $q_{e}$ and $q_{t}$ are the amount of toluene adsorbed (mg/g) at equilibrium and at time *t* (min), respectively.

The experimental data confirm that both the theoretical ($q_{e}$(th)) and experimental adsorption capacities ($q_{e}$(ex)) are in good agreement. The graph resulting from the pseudo-first-order expressively indicated that it does not fit linearly, which shows an irregular and nonlinear adsorption phenomenon [38].

The process of sorption is appropriately described by a pseudo-second-order kinetic model as it involves chemisorption [60], and these results of the kinetic models were duly supported by the outcomes of the Weber–Morris kinetic model for intra-particle diffusion, having a linear correlation of 0.998, as shown in Figure 11b. The intra-particle diffusion controls the sorption phenomenon.

### 3.4. Equilibrium Adsorption Study

Langmuir and Freundlich isotherm models were used to calculate the isotherm constants and the maximum adsorption capacities at various concentrations described in the Experimental Section to clearly understand the behavior of toluene.

Figure 13 is the linear plot of Ce/Qe as a function of Ce, where Ce is the equilibrium concentration of toluene while Qe is the amount of toluene at the adsorbent surface. The results indicated that the Langmuir isotherm was obeyed, which suggests that the adsorption is a monolayer and homogenous [61,62].

### 3.5. Sorption Mechanism of Toluene in Water

The proposed mechanism of toluene removal from water using the newly developed nanocomposite sorbent is presented in Figure 14. Based on the Raman, FTIR, and BET investigations, the proposed mechanism could be explained as follows:π–π interactions between CNT aromatic rings and/or polystyrene rings and the aromatic rings of toluene can occur.Hydrogen bonding could occur between the hydrogen atom from toluene and the oxygen atom from the metal oxides.CH–π interactions could occur between the hydrogen atom from toluene and the aromatic rings of the polystyrene or MWCNTs.

## 4. Conclusions

In the present work, it is clearly observed that the efficiency of MWCNTs increased when magnetite and polystyrene were added to prepare the PS:MWCNTs:Fe nano-sorbent for the removal of toluene from water. Characterization results (XRD, FTIR, RAMAN, SEM, TEM, and BET) were quite evident of the successful preparation of MWCNTs, which were then oxidized with a strong acid, followed by preparation of magnetite-MWCNTs composites after adding polystyrene and toluene. The results of the toluene adsorption experiments showed that the PS:MWCNTs:Fe nanocomposites achieved the highest removal efficiency and adsorption capacity of toluene removal compared with fresh, oxidized, and Fe-MWCNTs. The optimum parameters for toluene removal from water were: 60 min, 2 mg nano-sorbent dose, pH 5, solution temperature of 35 °C at 50 mL volume, toluene concentration of 50 mg/L, and shaking speed of 240 rpm. The adsorption kinetic study of toluene followed the pseudo-second-order kinetics with the best correlation, with an R^2^ value of 0.998, while the equilibrium adsorption study showed that the Langmuir isotherm was obeyed, which suggests that the adsorption is a monolayer and homogenous.

## Figures and Tables

**Figure 1 materials-14-05503-f001:**
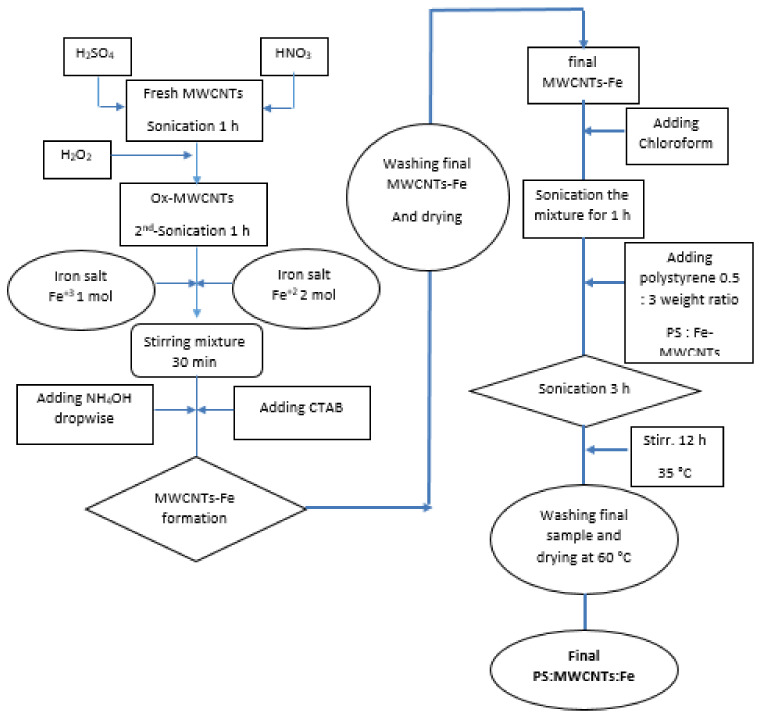
Schematic flow chart for the PS:MWCNTs:Fe nanocomposite preparation.

**Figure 2 materials-14-05503-f002:**
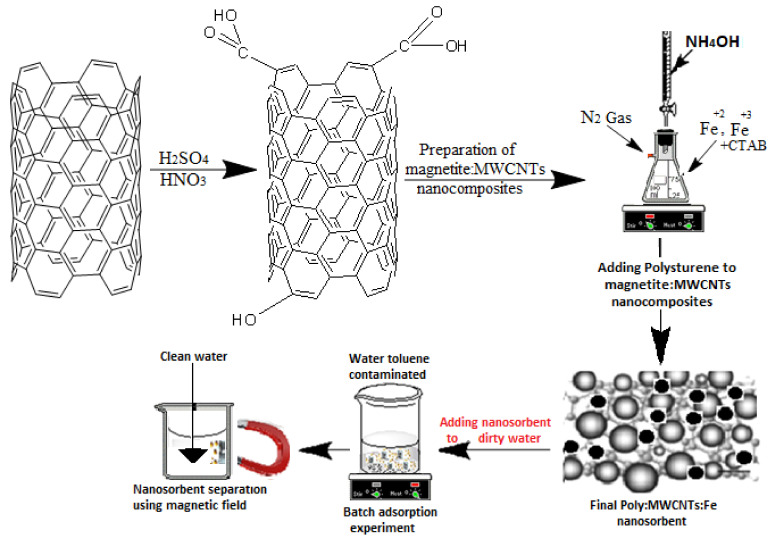
Schematic diagram for organic phase (toluene) removal in wastewater.

**Figure 3 materials-14-05503-f003:**
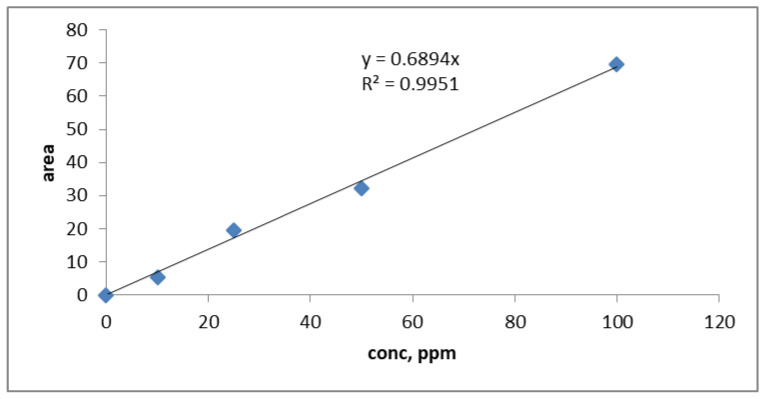
Standard calibration curve for toluene solutions performed by HPLC.

**Figure 4 materials-14-05503-f004:**
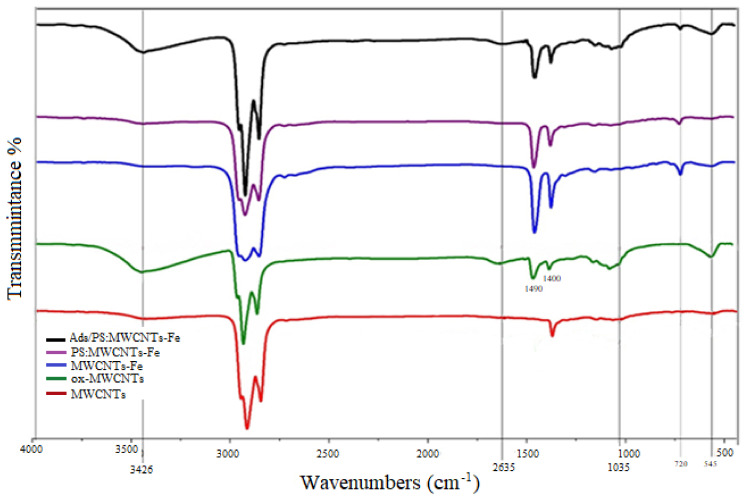
FTIR spectrum for fresh MWCNTs, ox-MWCNTs, Fe-MWCNTs, PS:MWCNTs:Fe, and PS:MWCNTs:Fe after toluene adsorption.

**Figure 5 materials-14-05503-f005:**
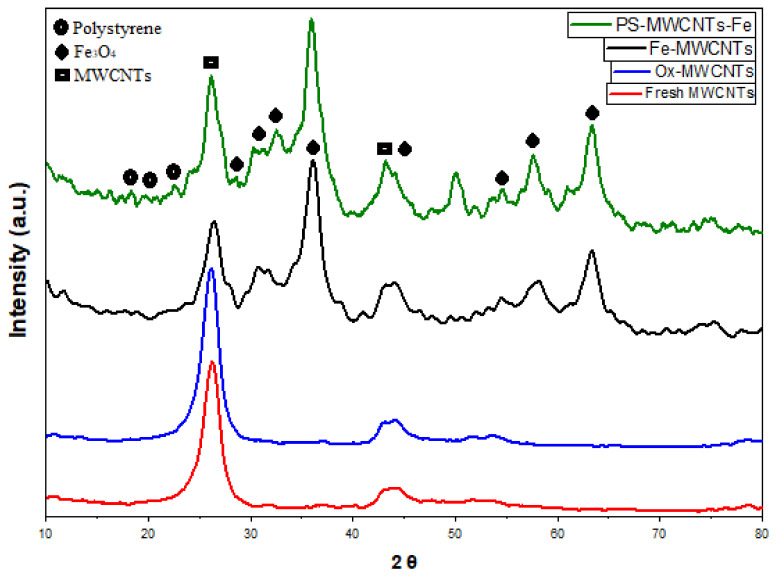
The XRD patterns for fresh MWCNTs, ox-MWCNTs, Fe-MWCNTs, and PS: MWCNTs:Fe nanocomposites.

**Figure 6 materials-14-05503-f006:**
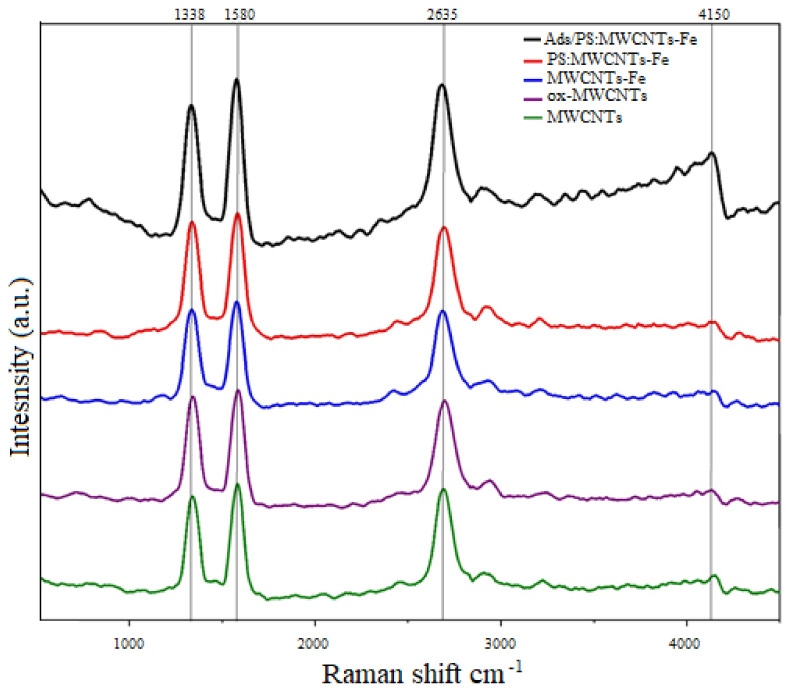
The Raman spectra for fresh MWCNTs, ox-MWCNTs, Fe-MWCNTs, PS:MWCNTs:Fe, and PS:MWCNTs:Fe after toluene adsorption.

**Figure 7 materials-14-05503-f007:**
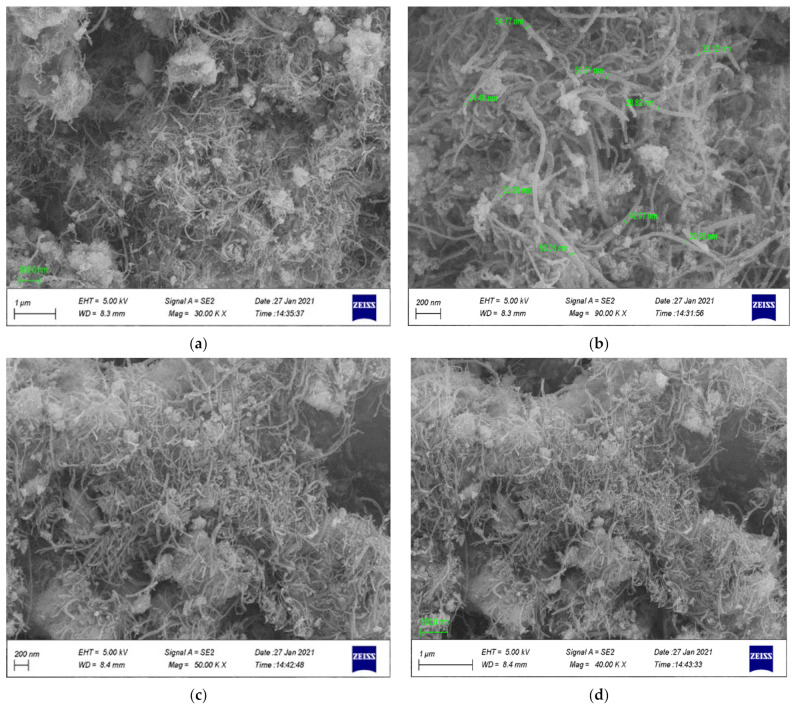
The SEM images for Fe:MWCNTs (**a**,**b**) and for PS:MWCNTs:Fe (**c**,**d**).

**Figure 8 materials-14-05503-f008:**
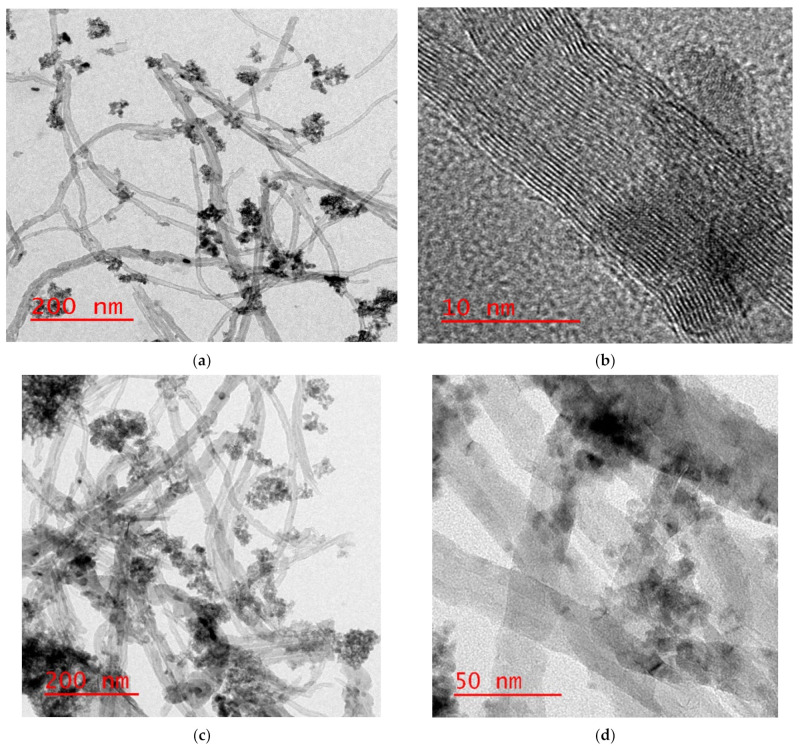
The TEM spectra for Fe:MWCNTs (**a**,**b**), PS:MWCNTs:Fe (**c**,**d**) and EDX results for Fe-MWCNTs (**e**).

**Figure 9 materials-14-05503-f009:**
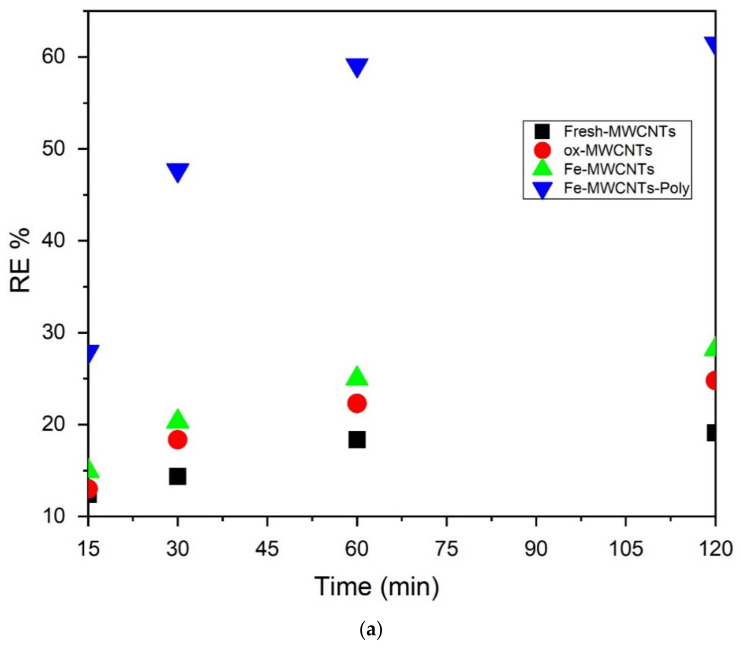
(**a**) Adsorption efficiency for removal of toluene from water as a function of time using fresh, ox-MWCNTs, Fe-MWCNTs, and PS-MWCNTs-Fe. (**b**) Decrease in the toluene concentration in water as a function of time using fresh, ox-MWCNTs, Fe-MWCNTs, and PS-MWCNTs-Fe.

**Figure 10 materials-14-05503-f010:**
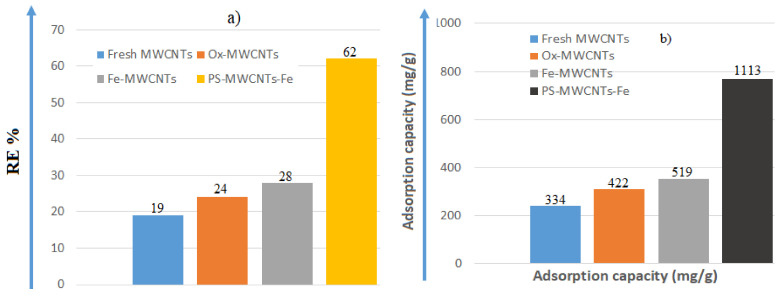
Flow chart explaining the removal efficiency (**a**) and adsorption capacity of nano-sorbents after 120 min (**b**).

**Figure 11 materials-14-05503-f011:**
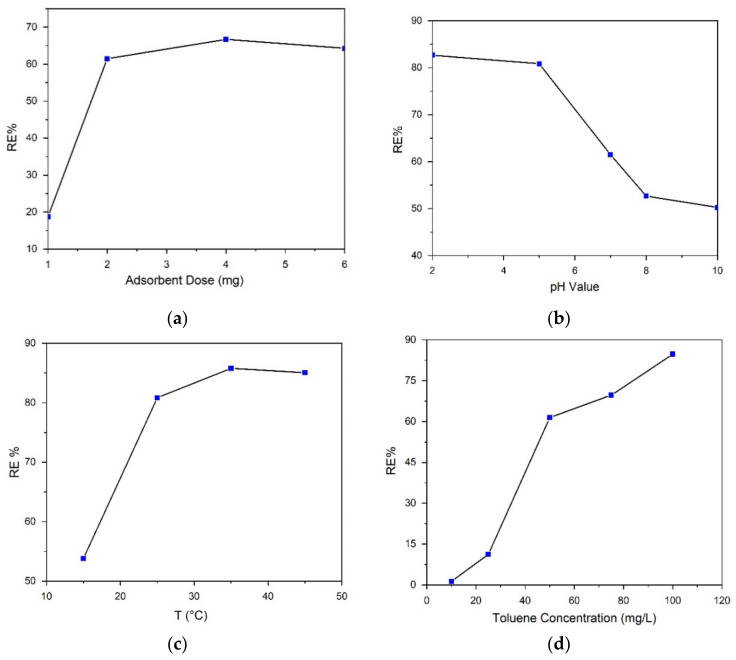
Effect of changing the adsorbent dosage (**a**), pH of the solution (**b**), temperature (**c**) and concentration of the solution (**d**) on the removal efficiency of toluene over the PS:MWCNTs:Fe nano-sorbent.

**Figure 12 materials-14-05503-f012:**
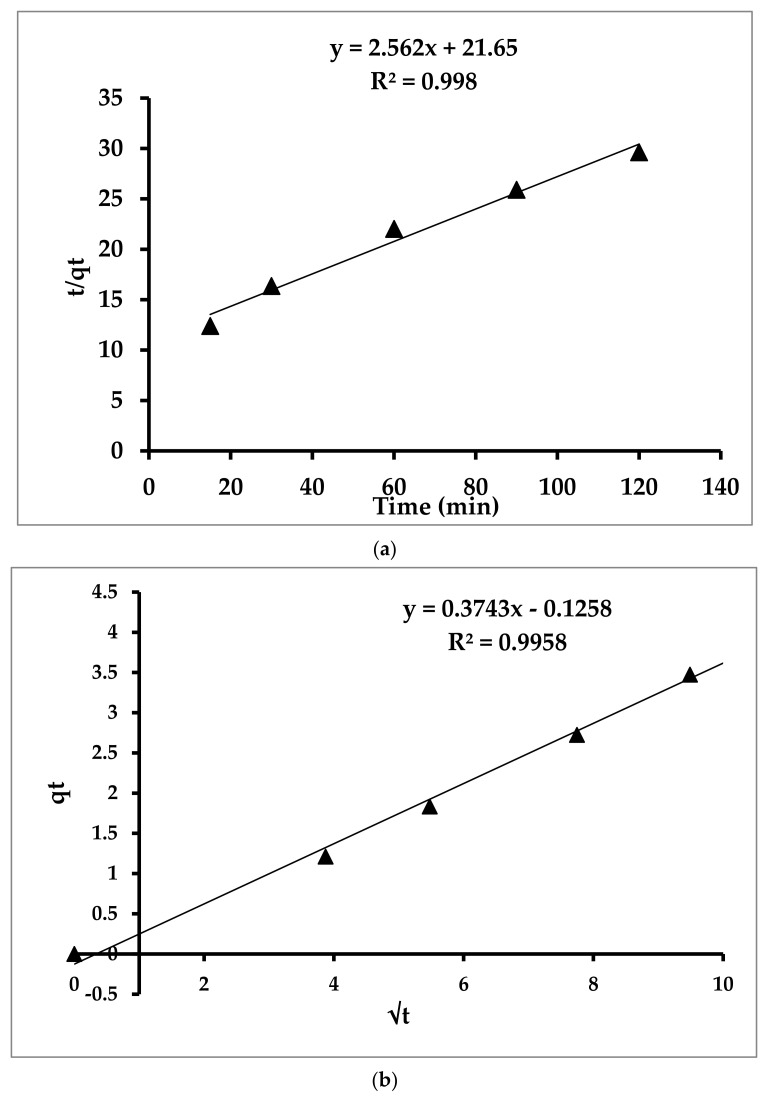
(**a**) Pseudo-second-order kinetic model for toluene adsorption. (**b**) Weber-Morris (intra-particle diffusion model) kinetic model for toluene adsorption.

**Figure 13 materials-14-05503-f013:**
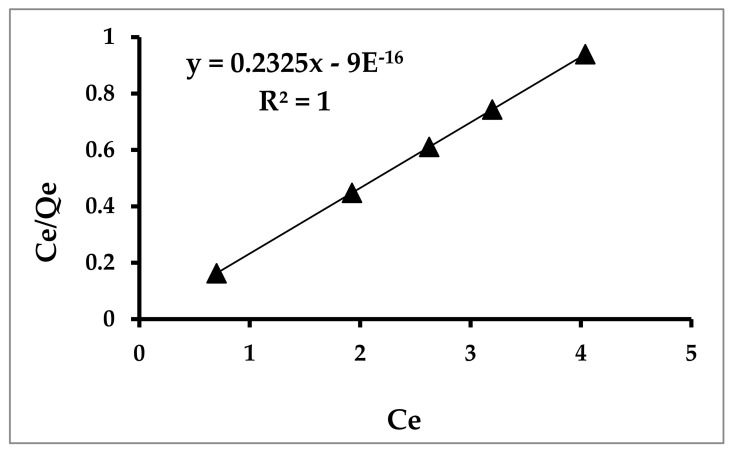
Langmuir isotherm model for toluene adsorption.

**Figure 14 materials-14-05503-f014:**
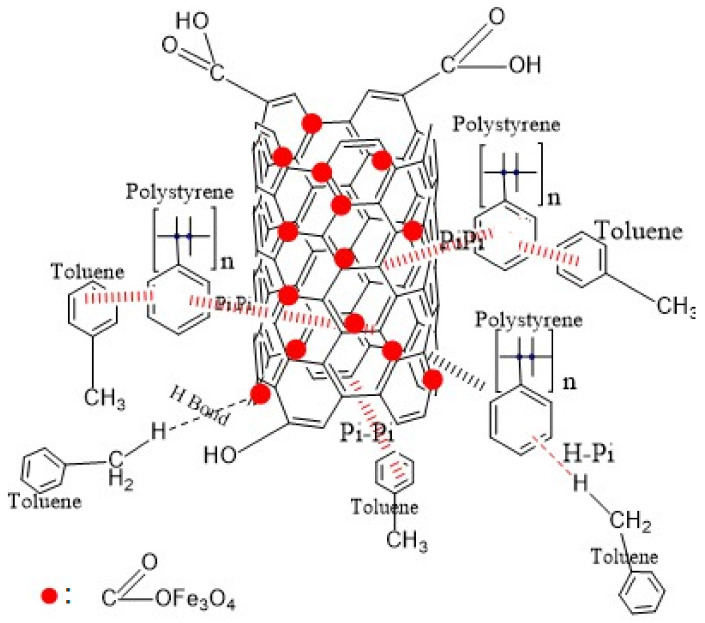
Proposed adsorption mechanism for toluene removal in water.

**Table 1 materials-14-05503-t001:** FTIR peak assignments for fresh and modified MWCNTs.

Compound	Vibrations (cm^−1^)	Explanation
Fresh MWCNTs	3425, 1625 cm^−1^	O–H Stretching and bending of water, respectively
	2925 cm^−1^	C–H Stretching of aromatic
	2845 cm^−1^	C–H Stretching of branching
	1460 cm^−1^	C=C Stretching of aromatic
	1395 cm^−1^	C–C Stretching of MWCNTs
Ox-MWCNTs	3420 cm^−1^	O–H Stretching of –COOH group
	1490 cm^−1^	C=O Stretching of –COOH group
	1400 cm^−1^	C–O Stretching of –COOH group
Fe-MWCNTs	720 cm^−1^	Fe–O Stretching of metal oxide with carboxyl group in MWCNTs
Ads/PS:MWCNTs:Fe	2925 cm^−1^	C–H Stretching of aromatic ring
	2845 cm^−1^	C–H Stretching of –CH3 group
	1035 cm^−1^	C–H bending in-plane
	545 cm^−1^	C–H bending in out-of-plane

**Table 2 materials-14-05503-t002:** Surface area, pore volume, and average pore size values of fresh and modified MWCNTs.

	Sample	S_BET_ m^2^/g	V_1.7–300 nm_, cm^3^/g	S_micro_ m^2^/g	V_micro_, cm^3^/g	D_av_, nm BJH
1	MWCNTs	161	0.7932	22.86	0.0096	17.9
2	Ox-MWCNTs	146	1.1142	13.9	0.0053	26.0
3	Fe_-_MWCNTs	233	0.5737	0	0	8.5
4	PS:MWCNTs:Fe	136	0.3372	0	0	8.5
5	PS:MWCNTs:Fe Ads.	129	0.3075	0	0	7.6

**Table 3 materials-14-05503-t003:** Adsorption capacities of different adsorbents for hydrocarbons.

Adsorbent Used	Oil Pollutant	Conditions	Dose	Capacity	Reference
			mg	(mg/g)	
Activated carbon	Toluene	100 mg/L, pH 6.9	10	501	[46]
GEL-SBA15	Toluene	650 mg/L, pH 7	10	597	[47]
CNTs-iron oxide	Toluene	100 mg/L, pH 7	50	381	[48]
CNTs-NaOCl	Toluene	20–200 mg/L, pH 7, T 25 °C	Oct-50	285	[49]
Modified activated carbon	Toluene	50 mg/L, pH 6, T 23 °C	50	126	[50]
Microemulsion/MWCNTs	Kerosene	500 mg/L, 25 °C	10	4700	[30]
KOH activated coconut shell based carbon treated with NH3	Toluene	50–250 mg/L, pH 6, T 30 °C, 115 rpm	100	357	[51]
Granule silica aerogel	Phenol	290 mg/L, pH 4–7, T 25 °C	500	142	[52]
Organo-clay CTMA	Toluene	0.0125–0.25 mg/L	200	58	[53]
LCNT-ox	Toluene	10–100 mg/L, pH 3–9, T 25 °C	20	72	[54]
Na-smectite	Toluene	100 mg/L, pH 6	-	410	[55]
Porous clay heterostructures (PCH)	Toluene	10–500 g/L, T 25 ± 2 °C, pH 3–11	0.5–4 g/L	101.1	[56]
High-performance activated carbon	Toluene	175–225 mg/L, T 25 °C, pH 7		1200	[57]
PS:MWCNTs-Fe	Toluene	200 mg/L, T 25 °C, pH 2–10	1 to 6	1113	Current work

## Data Availability

Not applicable.

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
