# Peer review of "Polystyrene-Fe3O4-MWCNTs Nanocomposites for Toluene Removal from Water"

_materials, 2021, doi:10.3390/ma14195503_

Round 1

Reviewer 1 Report

The authors have successfully developed nanocomposites suitable for water treatment as an efficient sorbent for oil pollutants. The most effective pollutant adsorbing nanocomposite was obtained by combining 10 nm sizes iron oxide nanoparticles with multi-walled carbon nanotubes and polystyrene (PS:MWCNTs:Fe) via a facile and higher scalable procedure. They characterized the structural and morphological properties by performing measurements with XRD, FTIR and Raman spectroscopy, SEM, and TEM, while they confirmed the sorption efficiency of this nanocomposite by measuring the BET surface area (136 m2/g) and adsorption capacity (200 mg kerosene per 1 L water). The manuscript is well written. The experimental data are carefully evaluated and consistently analyzed and clearly presented. I recommend that the manuscript be published as it is. 

Author Response

Thank you for your efforts to improve our article!

Please find the attached documents where we provided a point-by-point response to the your comments!

Reviewer 2 Report

This paper reports the preparation of polystyrene-magnetite functionalized multi-walled carbon nanotubes (MWCNTs) nanocomposites and their application in toluene removal from water. A wide range of characterization of the PS:MWCNTs:Fe nanomaterials has been carried out, including the FTIR, SEM, TEM, XRD and BET. They systematically study the structure, morphology, chemical nature, functional groups, and surface area properties of the prepared nano-adsorbents, and they further investigate the removal efficiency and adsorption capacity of toluene removal and also find the optimized parameters for the toluene removal of the PS:MWCNTs:Fe nanocomposites. I think these results are interesting and I recommend acceptance, subject to the following, minor corrections:

1 In the materials synthesis and preparation section (section 2.1), the authors report the use of a ratio of 1 : 3 between polystyrene and PS : Fe-MWCNTs. Are there any reasons why the authors decide to use this ratio? If yes, can they provide more literature for this?

2  In the FTIR characterization section (Figure 4), to provide a better comparison between these nanomaterials, can the authors provide the FTIR result of the polystyrene they use?

Author Response

(The authors gave the same response as above.)

Reviewer 3 Report

Review of paper no. materials-1271995 titled Polystyrene-Fe3O4-MWCNTs nanocomposites for toluene removal from water

This is an interesting and well-researched paper describing the synthesis and characterization of magnetic nanocomposites used to remove toluene from drinking water. The authors have demonstrated a very good adsorption efficiency (up to 70%). The paper is publishable subject to satisfactory authors’ responses to the following comments:

1.Figure 2 has a poor resolution. Please, improve the contrast of the schematic.

2.IR signals of Fresh MWCNTs and Oxidized MWCNTs should be interchanged in Fig. 4 to follow a natural order (NTs were first prepared and then oxidized). The same applies to Raman signals (Fig. 6).

3.Figure 7 should be split into 2 images (SEM and TEM images should be presented separately).

4.Parameters in Eq. (3) should be defined.

5.Last row in Table 2 summarizes the results of the present study. However, it mentions kerosene instead of toluene. Furthermore, Equations (1) and (2) also mention kerosene. Kerosene is a mixture of hydrocarbons. Toluene, on the other hand, is one specific hydrocarbon. Have you really used the kerosene? If so, what was the exact chemical composition of the mixture? How did you make sure that toluene was preferentially adsorbed and other hydrocarbons were left in the solution?

6.Table 2 (comparison with literature) needs to properly discussed.

7.Figure 13 shows possible adsorption sites for toluene on the magnetic nanocomposites. The figure needs a solid justification. Have you used a 1H/13C NMR spectroscopy to investigate the anticipated bonding?

8.English needs to be improved. The word “to be” is incorrectly used at many places in the manuscript (lines 387-392, 399-400, 461, and other places). The paper should be proof-read by native speaker before re-submission.

End of comments

Author Response

(The authors gave the same response as above.)

Reviewer 4 Report

The manuscript reported a novel design of nano adsorbents based on Polystyrene-Fe3O4-MWCNTs. The magnetic property of the nano adsorbents PS:MWCNTs: Fe was characterized, as well as the chemical nature and structure, morphology, functional groups, and surface area with a pore volume of the prepared nano adsorbents. PS:MWCNTs:Fe proved to achieve the highest removal efficiency and adsorption capacity of toluene removal compared to other samples prepared in this paper. The optimum parameters for toluene removal from water were also determined.

I consider the content of this manuscript will definitely meet the reading interests of the readers of the Materials journal as an environmentally friendly functional material. Although the design of experiments looks relatively monotonous, the overall characterization of this manuscript is still comprehensive and clear. Therefore, I suggest giving a major revision, by taking consideration of all the reviewers’ comments, the article should also have a lot of room for improvement. 

Also, I am afraid I have to point out that the English of the article needs to be polished and improved. In the minor issues, I tried to point out some English shortcomings, but there are still too many hidden corners to modify.

The details of my comments can be found in a separate word file.

Author Response

(The authors gave the same response as above.)

Round 2

Reviewer 3 Report

Authors answered most of my comments. The paper is acceptable for publication.

Reviewer 4 Report

The author made detailed replies to my previous comments. By reading the reply given by the author and comparing the corresponding changes before and after, I consider that the current version can be accepted, and I have no other suggestions for revision.